ecology/environmental science

Anthropocene, climate change, experimental design, ocean warming, ocean acidification, $CO_2$ emissions

**Author for correspondence:**
Nathan R. Geraldi
e-mail: nathan.geraldi@kaust.edu.sa

# A framework for experimental scenarios of global change in marine systems using coral reefs as a case study

Nathan R. Geraldi, Shannon G. Klein, Andrea Anton and Carlos M. Duarte

Institute of origin Red Sea Research Center (RSRC) and Computational Bioscience Research Center, King Abdullah University of Science and Technology (KAUST), Thuwal, Saudi Arabia

NRG, 0000-0002-2669-3867; SGK, 0000-0001-8190-3188; AA, 0000-0002-4104-2966

Understanding the consequences of rising $CO_2$ and warming on marine ecosystems is a pressing issue in ecology. Manipulative experiments that assess responses of biota to future ocean warming and acidification conditions form a necessary basis for expectations on how marine taxa may respond. Although designing experiments in the context of local variability is most appropriate, local temperature and $CO_2$ characteristics are often unknown as such measures necessitate significant resources, and even less is known about local future scenarios. To help address these issues, we summarize current uncertainties in $CO_2$ emission trajectories and climate sensitivity, examine region-specific changes in the ocean, and present a straightforward global framework to guide experimental designs. We advocate for the inclusion of multiple plausible future scenarios of predicted levels of ocean warming and acidification in forthcoming experimental research. Growing a robust experimental base is crucial to understanding the prospect form and function of marine ecosystems in the Anthropocene.

## 1. Introduction

Rising atmospheric $CO_2$ will continue to alter ecosystems worldwide through concomitant global warming and ocean acidification (OA) [1–3]. Although advances have been made in understanding the consequences of these anthropogenic drivers (e.g. [4–6]), our ability to anticipate the future of ecosystems requires quantifying responses to a palette of plausible future

**Figure 1.** (*a*) The projected greenhouse gas emissions through to the year 2100 based on the four representative concentration pathway scenarios (RCPs) (*b*) and the average temperature and atmospheric $CO_2$ concentrations (ppm). The coloured plume shows the spread of past and future projections from a hierarchy of climate carbon cycle models driven by historical emissions and the four RCPs to 2100. Ellipses shows global warming in 2100 versus cumulative $CO_2$ emissions from 1870 to 2100 from respective emission scenarios. The width of the ellipses in terms of temperature is caused by the impact of different scenarios for non-$CO_2$ climate drivers. The filled black ellipse shows observed emissions to 2005 and observed temperatures from 2000 to 2009 with associated uncertainties. Source: Ref: [7], fig. SPM.04 in Climate Change 2014: Synthesis Report.

climate scenarios. The selection of plausible scenarios for experimental research is complicated by spatial and temporal variation, uncertainties in future $CO_2$ emission trajectories and associated climate sensitivity (figure 1). Such uncertainties pose a substantial challenge for researchers who must inevitably simplify expected $CO_2$ concentrations and temperature in their experimental designs.

The consideration of local, baseline variation when determining ambient and future experimental levels is optimal [8–10]. However, local characteristics are often unknown probably because of the significant resources needed to measure them, particularly $CO_2$ and researchers should be aware of databases that collate relevant datasets (e.g. The Surface Ocean $CO_2$ Atlas and the Global $CO_2$ Time-Series and Moorings Project). The limited availability of local data and the need for a framework on how to choose experimental levels is highlighted by a review of temperature and $CO_2$ experiments along the west coast of the USA, which found that 80% and 13% of the studies gave no rationale for

temperature and $CO_2$ levels, respectively [8]. In addition, 45% of OA studies used mean surface global Intergovernmental Panel on Climate Change (IPCC) values for $CO_2$ levels, while 31% of experimental $CO_2$ levels were based on a combination of region models, local field data and IPCC projections [8]. Arguably, the need to understand the response of communities to environmental change is great enough to necessitate experiments even if local environmental characteristics are unknown. Although the IPCC assessments and the European Project on Ocean Acidification (EPOCA) report provide comprehensive information and future projections [7,11,12], we lack a parsimonious framework to guide scientists in the selection of experimental levels of projected $CO_2$ and warming. Here, we summarize current uncertainties in $CO_2$ emissions trajectories and provide a parsimonious framework that includes a comprehensive set of plausible $CO_2$ and warming scenarios, with the aim to aid in the design of climate change experiments when local characteristics and future projections are lacking.

## 2. Plausible future scenarios and associated uncertainties

The IPCC assessments provide estimates and associated uncertainties of future $CO_2$ concentrations and temperatures including representative concentration pathway scenarios (RCPs; figure 1). RCPs include an ambitious mitigation scenario (RCP2.6), two scenarios (RCP4.5 and 6.0) representing moderate reductions in $CO_2$ emissions [13] and a 'business-as-usual' scenario based on the absence of future efforts to reduce emissions (RCP8.5). $CO_2$ emissions and observed global warming, combined with projected trajectories in all RCPs towards the year 2100, depict a strong relationship between global cumulative $CO_2$ emissions and warming for both the global mean and for ocean surface temperature (figures 1b and 2a).

The most optimistic mitigation scenario (RCP2.6) relies on a rapid reduction in $CO_2$ emissions to reach net-zero greenhouse gas emissions (GHGs) towards the year 2080, in accordance with the Paris Agreement [7]. RCP2.6 restricts increases in atmospheric $CO_2$ concentrations to between 144 and 194 ppm and +1.6°C for the years 2081–2100 relative to pre-industrial values ([9], fig. 2.5b). Yet, recent assessments still estimate a median warming of +2.6 to +3°C, implying that a substantial (and unlikely) reduction in emissions is required to restrict warming to below +2°C [15]. For this reason, RCP4.5 and 6.0 (RCP6.0 equating to an increase of 351 ppm $CO_2$ and +2.8°C for the years 2081–2100 relative to pre-industrial values) are probable and warrant inclusion when selecting experimental treatments ([7], fig. 2.5b). Although efforts will hopefully be taken to reduce $CO_2$ emissions, the experimental evaluation of outcomes under the baseline scenario RCP8.5, corresponding to an increase of 614 ppm $CO_2$ and +4°C for the years 2081–2100 relative to pre-industrial values ([7], fig. 2.5b), is consistent with emission trajectories. Recent advances regarding uncertainties in climate sensitivity and concentrations of other GHGs suggest that ranges of $CO_2$ at +2°C may be underestimated by some commonly used models (e.g. CMIP5 ensemble), and that the 5 and 95 percentiles of current models for global increases in $CO_2$ concentrations at +2°C above pre-industrial levels are 143 and 820 ppm, respectively [16]. Given the uncertainty of atmospheric $CO_2$ trajectories (figure 1), we suggest there is a clear need to explore the ecological consequences of all RCPs.

A potential complication for marine scientists is that the majority of the IPCC data provides global surface projections, but the ocean is warming slower than land [17] (figure 2a). Within the IPCC reports, projections of $CO_2$ content and warming for the global ocean are limited relative to those that apply to the atmosphere. However, we extracted the mean temperature projections for the four RCPs towards the end of this century based on the mean of the three main ocean basins ([7], table SM30–4; figure 1a). Future oceanic $CO_2$ values are needed given that marine values often deviate from atmospheric $CO_2$ values [18] (figure 2a). The IPCC's regional projections exclude polar areas, which is notable given that the Arctic is warming two to three times faster than the global average [19,20]. In this study (and from now on), we focus on the marine-only projections, unless otherwise noted.

## 3. Framework for designing research on future warming and elevated $CO_2$

We present a framework for designing experiments to assess the responses of marine biota to future climate scenarios that encompasses a range of scenarios considered in the IPCC projections (figures 1 and 2a). As a case study, we use the database of experimental warming and OA experiments for coral reef ecosystems provided by Hughes et al. [14] to compare the temperature and $pCO_2$ manipulations reported in the literature to the range of possible scenarios (figure 2c,d). In this case, approximately one-fourth of warming levels employed (24%; figure 2b) fell within scenarios (less than or equal to 2.79°C, RCP8.5). We based this assessment on levels of ocean warming expected from recent temperatures (i.e. ambient,

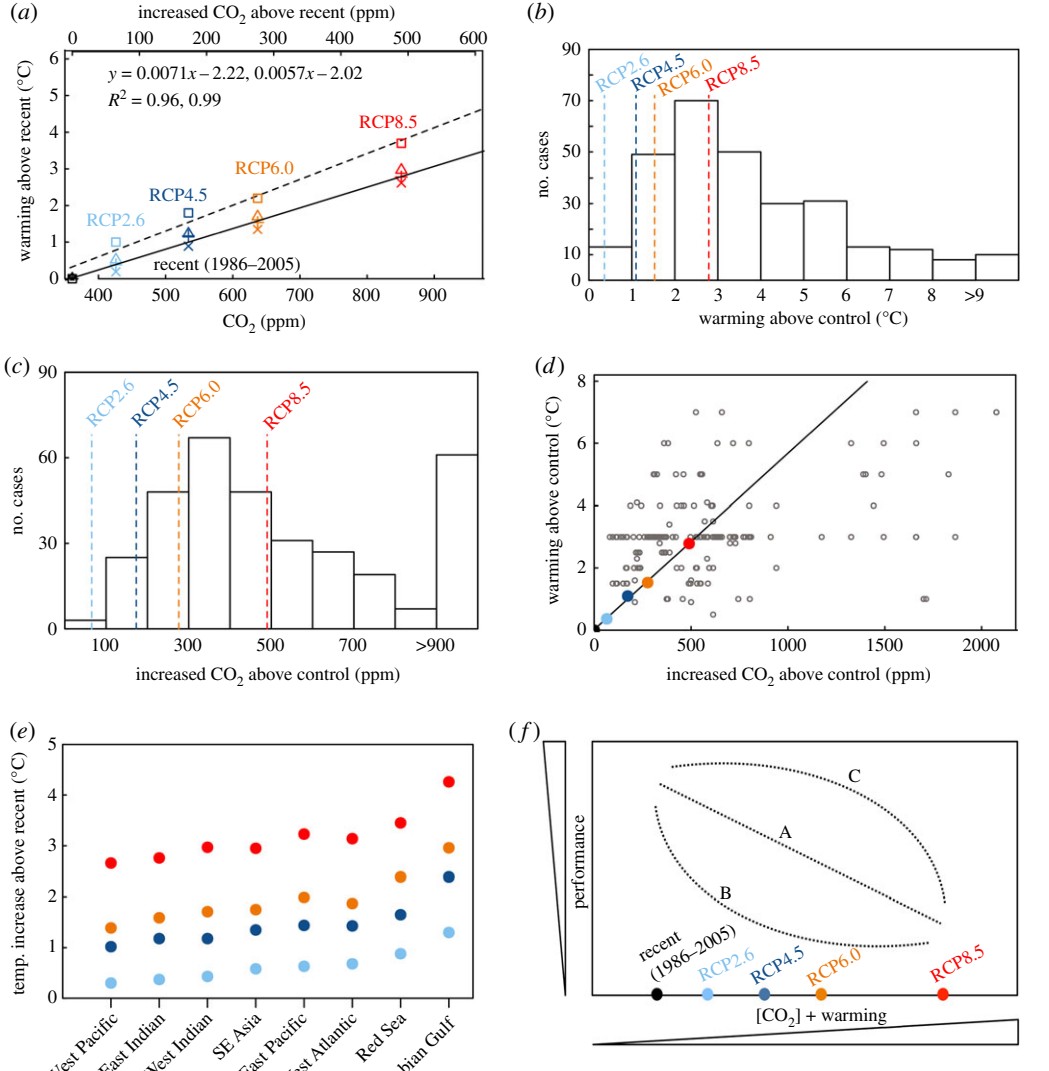

**Figure 2.** Climate change and experiments. (*a*) The linear relationship between projected global atmospheric $CO_2$ concentrations (squares and dashed line) and surface temperatures of the world (solid line) and oceans (Atlantic, triangles; Pacific, crosses; Indian, x marks). Data for panel (*a*) was extracted from fig. 2.5b and table SM30-4 of SPM, IPCC report [7]. Symbols indicate projected values for the end of the twenty-first century under each RCP. The frequency of (*b*) temperature and (*c*) $CO_2$ levels used to experimentally simulate warming and acidification in coral reef research. Data for panels (*b*,*c*) were obtained from the electronic supplementary material, tables S1 and S2 of [12] and data we added (RCP overlay assumes control of recent $CO_2$, 361 ppm; electronic supplementary material, S4). (*d*) Relationship between levels of warming and elevated $CO_2$ within dual-stressor treatments of coral reef research [14], and the linear relationship between projected oceanic $CO_2$ concentrations and increases in ocean temperature from (*a*). (*e*) Region-specific temperature increases expected for eight major coral reef provinces for the years 2010–2099 under RCPs ([7], table SM30-4, Ch. 30SM). (*f*) Theoretical performance of marine organisms to an experimental gradient of dual climate change stressors ($CO_2$ concentrations and warming) using a continuous scale, where A represents a linear decrease in performance, while B and C represent two of many potential nonlinear responses. The diagram in (*f*) represents a negative effect of stressors, but both null and positive effects are also possible. Colours in all panels represent values projected (surface ocean mean for *b*–*d*) under RCP2.6 (light blue), RCP4.5 (dark blue), RCP6.0 (orange) and RCP8.5 (red). Data is provided in the electronic supplementary material, S1–S4.

current-day conditions) as most studies in this dataset were conducted post year 2000. However, Hughes *et al.* [14] assessed whether warming levels in these studies aligned with those expected from pre-industrial temperatures levels, highlighting the need to distinguish between increases from current conditions versus pre-industrial levels in experimental studies. To assess the difference between experimental and control *p*$CO_2$, we revisited references provided by Hughes *et al.* as their data only included levels for future treatments (control *p*$CO_2$ is included in the electronic supplementary material, S4). Approximately 57% of the studies used elevated *p*$CO_2$ concentrations that were within RCP scenarios (less than or equal to +490 ppm, RCP8.5; figure 2*c*). This suggests that the majority of studies are assessing impacts within

expected scenarios, although there remain many studies (43%) that may be overestimating the consequences of OA. For other marine biomes, it is probable that experimental designs may also require prompt assessment, and future reviews that quantify experimental treatments are warranted.

Projected increases in $CO_2$ and temperature are correlated at broad-scales (figures 1$b$ and 2$a$). However, the majority of coral reef studies from Hughes $et\ al.$ [14] (62%) that aimed to simulate future ocean warming and OA manipulated these drivers independently (figure 2$d$). Of the remaining studies that assessed the drivers concomitantly (38%), most levels of warming and elevated $p$CO$_2$ within dual-stressor treatments deviate from the linear relationship between $CO_2$ concentrations and warming (figure 2$d$). Although this may reflect variability associated with local characteristics, existing reviews indicate the majority of studies do not base experimental levels on local conditions [8]. Ecologists may consider shifting their experimental designs to a gradient approach that explores a range of $CO_2$ and warming conditions given that responses are possibly nonlinear [8,12,18,21–24] (figure 2$f$). Nonlinear responses could also result in null effects or positive effects (not shown in figure 2$f$). Theoretical predictions have been made to estimate the nature of biotic responses to the dual stressors along a continuous gradient [25,26] and although full factorial experiments remain critical [27,28], experimental data of biota responses over a continuous scale of climate change scenarios are needed.

We recognize our proposed framework probably oversimplifies the $CO_2$ and temperature regimes that vary locally, but accurate characterizations of local and regional variability are currently rare, especially for $p$CO$_2$ concentrations [29]. As we move forward to characterize and understand drives of high-frequency temperature and $CO_2$ regimes in coastal systems, this framework could be used to complement baseline observations. This is especially vital for future research focusing on coastal marine habitats that already experience temperatures or $p$CO$_2$ levels considerably higher than large-scale means or future projections. Large local variation can result from $in\ situ$ biological processes [30–33], watershed characteristics [34–36] and upwelling [37].

# 4. Regional-specific climate change

Global projections of OA and warming may not represent specific systems and choosing levels for experimental research warrants consideration of projections specific to the geographical location being studied [38]. For instance, global average temperature estimates can be greater than future temperature increases in the ocean as well as in specific ocean provinces, and experiments should account for region-specific heterogeneity [14] which is summarized for eight major coral reef provinces (figure 2$e$). Fine-scale projections of future temperatures for each RCP are also available as global layers (approx. 10 km grid of globe, http://www.bio-oracle.org/) [39]. Measures and projections of local $CO_2$ regimes are scarce because $p$CO$_2$ levels vary considerably from atmosphere levels because of community metabolism [40], local geology [36] or upwelling [37], and researchers often need find an alternative method to determine experimental levels. $In\ lieu$ of such information, our framework could be used to obtain a proxy of $\Delta CO_2$ based on the linear relationship with temperature (figure 2$a$). For example, in the case of coral reef provinces, the possible range of end-of-century warming and $\Delta CO_2$ that would need to be explored spans from 1.5°C to 4.5°C (relative to pre-industrial) and from 144 to 614 ppm, respectively (figure 2$d$). The IPCC provides comprehensive temperature projections for the near- (years 2010–2039) and long-term (years 2010–2099) scenarios for most marine regions ([7], table SM30–4, Ch. 30SM).

# 5. Comparison to published experimental suggestions

The EPOCA [11,12] suggested several levels of $CO_2$ for the design of experiments testing OA. They suggest 280 ppm (pre-industrial), 385 ppm (present day), 750 ppm (moderate prediction) and then include 1000 ppm (high prediction) and more increments in between these values if possible [12]. We make similar suggestions based on different RCPs (global mean), which includes 360, 430, 530, 640 and 850 ppm, corresponding to recent (1986–2005), and RCPs 2.6, 4.5, 6.0 and 8.5. We provide suggestions on how to manipulate concomitant temperature and $CO_2$, which is not provided by EPOCA but highly relevant under current and future climate change conditions [9,10].

Few reviews have assessed whether experimental treatments are tailored to plausible future climate conditions. Exceptions include a review of empirical studies that simulated global warming and OA on coral reef organisms [14] and on species in upwelling coastal systems along the USA west coast [8]. Hughes $et\ al.$'s [14] recommendations for forthcoming experiments of warming and OA focused on

the global surface mean (both land and oceans) that relied on a rapid transition to net-zero GHGs [41] and restraining global warming to less than +2°C (approx. 410–420 ppm atmospheric $CO_2$) [14]. These calculations were based on equilibrium climate sensitivity [41], which is generally intended as benchmarks for comparing the magnitude of climate response projected by climate models [42]. We advocate for the preferential use of RCPs, as adopted by the IPCC [7], for estimating future $CO_2$ concentrations and warming. IPCC scenarios indicate that a +2°C (1.4 above current 1986–2005 levels) increase in global mean temperature (relative to pre-industrial) corresponds with a mean increase in atmospheric $CO_2$ concentration of +234 ppm to approximately 520 ppm (figure 2a), some 100 ppm greater than suggestion by Hughes *et al.* [14]. If focusing on marines systems, a +1.4°C increase above current levels corresponds with an atmospheric concentration of approximately 705 ppm (figure 2a). Reum *et al.* [8] provide two insightful frameworks for determining levels of experiments that manipulate temperature and $CO_2$. The first used three temperature levels based on local measures and two $CO_2$ levels based on IPCC future ocean surface $CO_2$ (390 and 788 ppm). The second uses local measures of both temperature and $CO_2$ with future $CO_2$ levels based on present-day local measures and future dissolved inorganic carbon estimates. Their framework is very useful when local characteristics are available. However, some researchers will need to determine levels for experiments manipulating temperature and $CO_2$ when knowledge of local characteristics is lacking. Our framework provides a starting point and location of pertinent information.

## 6. Baselines in climate change experiments

A reoccurring issue, which seems to be overlooked by many climate change researchers, is whether experimental manipulations are based on increases from pre-industrial or present conditions. For instance, the degree of warming projected in the RCPs are typically values relative to the years 1850–1900, and given that the globe (on average) has already warmed by approximately 0.88°C, +2.4°C warming projected in RCP4.5 would equate to a global increase of approximately 1.52°C from current conditions. What might come across as an obvious and simple concept, may be unnoticed in the manipulation of experimental treatment levels for warming where researchers apply levels of warming projected in RCPs to current conditions, inadvertently treating ambient conditions as those of the pre-industrial era.

## 7. Conclusion

Given that time series of local temperature and $CO_2$ concentration are often lacking and the substantial uncertainties in future projections and climate sensitivity, we propose that a slate of likely climate change scenarios need to be explored in experiments to provide a 'covering all bases' approach to understand future marine ecosystems. Although we primarily focused on the ocean, much of the discussion and framework could also apply to terrestrial and freshwater ecosystems. Conducting experiments that replicate local, baseline variation alongside future scenarios necessitates complicated logistical efforts and significant resource investments, especially when dealing with $CO_2$. We hope recommendations provided here enhance the accuracy of future studies and initiate discussion among researchers to improve the exploration of the future performance of biota in the Anthropocene.

Data accessibility. Data for this article are in the electronic supplementary material.
Authors' contributions. N.R.G., S.G.K., A.A. and C.M.D. conceived the study. N.R.G., S.G.K. and A.A. collated data and N.R.G. created the figure. N.R.G. and S.G.K. wrote the initial draft of the manuscript, while all authors contributed to editing of the manuscript. All authors gave final approval for publication.
Competing interests. Authors have no competing interests.
Funding. Funding provided by KAUST.
Acknowledgements. We thank C. Brown, D. J. Suggett and an editor from Biology Letters for comments.

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
