## [Reviewer comments · Royal Society Open Science]

Review History

RSOS-191118.R0 (Original submission)

Review form: Reviewer 1

Is the manuscript scientifically sound in its present form?

No

Are the interpretations and conclusions justified by the results?

No

Is the language acceptable?

Yes

Do you have any ethical concerns with this paper?

No

Have you any concerns about statistical analyses in this paper?

No

Recommendation?

Accept with minor revision (please list in comments)

Comments to the Author(s)

I thank the authors for their thoughtful responses and considerations of the prior review. I believe that the authors addressed most of the major concerns by the prior review.

My only major critiques are on figure 2 and the acclimatization section.

First, the authors' response to the reviewer comment on figure 2 was:

"We agree with the reviewer 100% that the relative CO₂ change would be more appropriate and that this varies greatly. However, most studies do not give relative CO₂, but give absolute values. This highlights the issue that most researchers do not know the local CO₂ conditions and variability."

Each of these experiments usually have a "control", "ambient", or "current day" scenario. Even if they report in absolute values the experimental design is that the treatment, CO₂ values are relative to a control. A plot showing delta CO₂ (treatment - control/ambient) will likely get rid of some of this noise, whether or not they know the local variability of the system (in the same way that temperature is presented).

Second, figure 2f assumes that all marine organisms are "losers" of climate change (linear and non-linear responses are all negative and the legend says "marine organisms"). This is an overgeneralization of performance of marine organisms as some species have no or a positive response in different performance measures (see, Kroeker et al. 2013).

The "organism acclimatization" section. I completely agree with the authors that understanding how/if organisms can acclimatize to OA/warming is important. However, as currently written, it seems like an afterthought to the main goal of the paper. If acclimatization is brought into the framework then it should be accompanied by suggestions for timing for experiments, etc (as rates of change are brought up in this paragraph). I imagine this is beyond the scope of this study, so I would either remove this section or talk more broadly about how this framework will help us understand responses of marine organisms to climate change. For example, why is acclimatization called out within the context of the framework presented and not species interactions, ecosystem responses, etc.?

Typo on Line 38: change Rational to rationale

Review form: Reviewer 2

Is the manuscript scientifically sound in its present form?

Yes

Are the interpretations and conclusions justified by the results?

Yes

Is the language acceptable?

Yes

Do you have any ethical concerns with this paper?

No

Have you any concerns about statistical analyses in this paper?

No

Recommendation?

Major revision is needed (please make suggestions in comments)

Comments to the Author(s)

This paper is an interesting examination of experimental conditions used for temperature and CO₂ experiments for corals. While an important topic I feel is either pitched too low for people already in the field who are conducting studies or misses a number of points for those entering the field. The study also duplicates some areas that have been discussed in other papers, e.g. Hughes et al.

Specific comments

The paper quite rightly advocates for experiments to be put into the context of their local environment, and examines how different oceanic basins are being differentially affected but I think misses variation below this scale and issues with down scaling climate models. For example Kwiatkowski et al. (Climate Dynamics September 2014, Volume 43, Issue 5–6, pp 1483–1496 | Cite as

What spatial scales are believable for climate model projections of sea surface temperature?) specifically examines the difficulties of using global models to project local (even regional) scale future scenarios. Even smaller than this there is little discussion of the need to include daily variations (this appears to have been brought up in a previous revision but should be further expanded upon). The paper needs to address what scale of local CO₂ and temperature variation is seen, particularly daily changes. In some places temperature can vary by degrees and pCO₂ levels can change over 200 ppm

It is useful to set out best practice but there should also be an acknowledgement of the difficulty in conducting experiments of this type. For example it is not experimentally easy to match temperature and CO₂ conditions, as stated above should this include daily variation. Also the acclimation period. I think all experimentalists would like to acclimate their corals for years but this is logistically almost impossible.

The paper relied heavily on Hughes et al 2017 for their summary of experimental studies, firstly this summary is not complete as it is missing a number of studies, secondly that is now two years old so I think it is worthwhile including studies from the last two years.

You need a citation for the sentence, or the analysis in the supplements, at line 36 which examines studies on the US west coast.

Why is there not RCP2.6 in Figure 2E

Decision letter (RSOS-191118.R0)

08-Sep-2019

Dear Dr Gerald,

The editors assigned to your paper ("A framework for experimental scenarios of global change in marine systems using coral reefs as a case study.") have now received comments from reviewers. We would like you to revise your paper in accordance with the referee and Associate Editor suggestions which can be found below (not including confidential reports to the Editor). Please note this decision does not guarantee eventual acceptance.

Please submit a copy of your revised paper before 01-Oct-2019. Please note that the revision deadline will expire at 00.00am on this date. If we do not hear from you within this time then it will be assumed that the paper has been withdrawn. In exceptional circumstances, extensions may be possible if agreed with the Editorial Office in advance. We do not allow multiple rounds of revision so we urge you to make every effort to fully address all of the comments at this stage. If deemed necessary by the Editors, your manuscript will be sent back to one or more of the original reviewers for assessment. If the original reviewers are not available, we may invite new reviewers.

- Data accessibility

If you wish to submit your supporting data or code to Dryad (<http://datadryad.org/>), or modify your current submission to dryad, please use the following link:
<http://datadryad.org/submit?journalID=RSOS&manu=RSOS-191118>

- **Competing interests**

- **Authors' contributions**

- **Acknowledgements**

- **Funding statement**

Kind regards,

Royal Society Open Science Editorial Office
Royal Society Open Science
openscience@royalsociety.org

on behalf of Prof Kevin Padian (Subject Editor)
openscience@royalsociety.org

Associate Editor's comments:

Thank you for submitting the manuscript for consideration following transfer. In addition to an original reviewer from the earlier submission, the journal has sought advice from a second referee. Both see merit in publishing your work, and in the former's case recognise the efforts you have made to address their earlier concerns. However, a number of critiques remain that require your attention. Please address these in your revision, along with a point-by-point reply to the reviewers (this helps both Editors and reviewers enormously should further review be required).

Comments to Author:

Reviewers' Comments to Author:

Reviewer: 1

Comments to the Author(s)

I thank the authors for their thoughtful responses and considerations of the prior review. I believe that the authors addressed most of the major concerns by the prior review.

My only major critiques are on figure 2 and the acclimatization section.

First, the authors' response to the reviewer comment on figure 2 was:

"We agree with the reviewer 100% that the relative CO₂ change would be more appropriate and that this varies greatly. However, most studies do not give relative CO₂, but give absolute values. This highlights the issue that most researchers do not know the local CO₂ conditions and variability."

Each of these experiments usually have a "control", "ambient", or "current day" scenario. Even if they report in absolute values the experimental design is that the treatment, CO₂ values are relative to a control. A plot showing delta CO₂ (treatment - control/ambient) will likely get rid of some of this noise, whether or not they know the local variability of the system (in the same way that temperature is presented).

Second, figure 2f assumes that all marine organisms are "losers" of climate change (linear and non-linear responses are all negative and the legend says "marine organisms"). This is an overgeneralization of performance of marine organisms as some species have no or a positive response in different performance measures (see, Kroeker et al. 2013).

The "organism acclimatization" section. I completely agree with the authors that understanding how/if organisms can acclimatize to OA/warming is important. However, as currently written, it seems like an afterthought to the main goal of the paper. If acclimatization is brought into the framework then it should be accompanied by suggestions for timing for experiments, etc (as rates of change are brought up in this paragraph). I imagine this is beyond the scope of this study, so I would either remove this section or talk more broadly about how this framework will help us understand responses of marine organisms to climate change. For example, why is acclimatization called out within the context of the framework presented and not species interactions, ecosystem responses, etc.?

Typo on Line 38: change Rational to rationale

Reviewer: 2

Comments to the Author(s)

This paper is an interesting examination of experimental conditions used for temperature and CO₂ experiments for corals. While an important topic I feel is either pitched too low for people already in the field who are conducting studies or misses a number of points for those entering the field. The study also duplicates some areas that have been discussed in other papers, e.g. Hughes et al.

Specific comments

The paper quite rightly advocates for experiments to be put into the context of their local environment, and examines how different oceanic basins are being differentially effected but I

think misses variation below this scale and issues with down scaling climate models. For example Kwiatkowski et al. (Climate Dynamics September 2014, Volume 43, Issue 5–6, pp 1483–1496 | Cite as What spatial scales are believable for climate model projections of sea surface temperature?) specifically examines the difficulties or using global models to project local (even regional) scale future scenarios. Even smaller than this there is little discussion of the need to include daily variations (this appears to have been brought up in a previous revision but should be further expanded upon). The paper needs to address what scale of local CO₂ and temperature variation is seen, particularly daily changes. In some place temperature can vary but 6y degrees and pCO₂ levels can change over 200 ppm

It is useful to set out best practice but there should also be an acknowledgement of the difficulty in conducting experiments experiments of this type. For example it is not experimentally easy to match temperature and CO₂ conditions, as stated above should this include daily variation. Also the acclimation period. I think all experimentalist would like to acclimate their corals for years but this is logistilcally almost impossible.

The paper realised heavily on Hughes et al 2017 for their summary of experimental studies, Firstly this summary is not complete as is missing a number of studies, secondly that is now two years old so I think it is worthwhile including studies from the last two years.

You need a citation for the sentence, or the analysis in the supps, at line 36 which examines studies on the US west coast.

Why is there not RCP2.6 in Figure 2E

Author's Response to Decision Letter for (RSOS-191118.R0)

See Appendix A.

RSOS-191118.R1 (Revision)

Review form: Reviewer 1

Is the manuscript scientifically sound in its present form?

No

Are the interpretations and conclusions justified by the results?

Yes

Is the language acceptable?

Yes

Do you have any ethical concerns with this paper?

No

Have you any concerns about statistical analyses in this paper?

No

Recommendation?

Accept with minor revision (please list in comments)

Comments to the Author(s)

Thank you for your careful revisions on the last draft. I have a few more minor suggestions. I have been meticulous on these reviews because I think that suggesting a framework for all scientists to follow is something that should not be taken lightly.

Line 137: add the majority of coral reef studies “from Hughes et al”. These aren’t the majority of coral reef studies, they are the majority from a recent meta-analysis.

Line 152: I am still really having a hard time with ignoring local variability in CO₂/temp in your framework. As the other reviewer stated, local diel variability can be extreme which will have a substantial impact on biological responses. CO₂ is also so different from site to site and even region to region. Even if you don’t have the capability to measure the seawater of the location where you are collecting your biological samples (though this does seem unlikely if you are able to measure the water is an experimental aquarium), then at the very least you can use the extensive data on global ocean CO₂ which has seasonal data by region. I think incorporating this information into your framework is important. Here are a few examples of global ocean CO₂ datasets that are publicly available. Each of these have regional data, and most also have at least seasonal or monthly data.

https://www.nodc.noaa.gov/ocads/oceans/ndp_094/ndp094.html

<https://www.socat.info/>

<https://www.nodc.noaa.gov/ocads/oceans/>

Line 158: There are tons of studies on “in situ biological processes” affecting local CO₂ variation in addition to [31] including, but not limited to:

Kleypas, J. A., Anthony, K. & Gattuso, J. P. Coral reefs modify their seawater carbon chemistry—case study from a barrier reef (Moorea, French Polynesia). *Global Change Biology* 17, 3667–3678 (2011).

DeCarlo, T. M. et al. Community production modulates coral reef pH and the sensitivity of ecosystem calcification to ocean acidification. *Journal of Geophysical Research: Oceans* 122, 745–761 (2017).

Silbiger N.J and Sorte C.J.B. Biophysical feedbacks mediate carbonate chemistry on coastal ecosystems across spatiotemporal gradients *Scientific Reports* 796, (2018).

Page, H. N. et al. Differential modification of seawater carbonate chemistry by major coral reef benthic communities. *Coral Reefs* 35, 1311–1325 (2016).

Smith, J. E., Price, N. N., Nelson, C. E. & Haas, A. F. Coupled changes in oxygen concentration and pH caused by metabolism of benthic coral reef organisms. *Marine Biology* 160, 2437–2447 (2013).

Kapsenberg, L. & Hofmann, G. E. Ocean pH time-series and drivers of variability along the northern Channel Islands, California, USA. *Limnology and Oceanography* 61, 953–968 (2016).

Yeakel, K. L. et al. Shifts in coral reef biogeochemistry and resulting acidification linked to offshore productivity. *Proceedings of the National Academy of Sciences* 112, 14512–14517 (2015).

Krause-Jensen, D. et al. Macroalgae contribute to nested mosaics of pH variability in a subarctic fjord. *Biogeosciences* 12, 4895–4911 (2015).

Similarly, there are many many studies on watershed characteristics and upwelling with respect to affecting carbonate chemistry/temperature regimes. I would add more citations to these statements to highlight the substantial amount of work put into these disciplines.

Units are missing from some of the supplemental files

Review form: Reviewer 2

Is the manuscript scientifically sound in its present form?

Yes

Are the interpretations and conclusions justified by the results?

Yes

Is the language acceptable?

Yes

Do you have any ethical concerns with this paper?

No

Have you any concerns about statistical analyses in this paper?

No

Recommendation?

Accept as is

Comments to the Author(s)

The authors have addressed those issues I have previously brought up, and I think the paper will be a valuable addition to the field.

Decision letter (RSOS-191118.R1)

04-Nov-2019

Dear Dr Geraldi:

On behalf of the Editors, I am pleased to inform you that your Manuscript RSOS-191118.R1 entitled "A framework for experimental scenarios of global change in marine systems using coral reefs as a case study." has been accepted for publication in *Royal Society Open Science* subject to

minor revision in accordance with the referee suggestions. Please find the referees' comments at the end of this email.

The reviewers and Subject Editor have recommended publication, but also suggest some minor revisions to your manuscript. Therefore, I invite you to respond to the comments and revise your manuscript.

- Ethics statement

- Data accessibility

If you wish to submit your supporting data or code to Dryad (<http://datadryad.org/>), or modify your current submission to dryad, please use the following link:
<http://datadryad.org/submit?journalID=RSOS&manu=RSOS-191118.R1>

- Competing interests

- Authors' contributions

- Acknowledgements

- Funding statement

Because the schedule for publication is very tight, it is a condition of publication that you submit the revised version of your manuscript before 13-Nov-2019. Please note that the revision deadline will expire at 00.00am on this date. If you do not think you will be able to meet this date please let me know immediately.

on behalf of Prof Kevin Padian (Subject Editor)
openscience@royalsociety.org

Associate Editor Comments to Author:

Thank you for submitting your revised paper to us, and for providing such a thorough response to the previous comments made. We are pleased to accept this manuscript in principle, owing to the remaining comments being addressed. As you can see, the reviewers are now mostly satisfied, however Reviewer 2 has some remaining concerns and suggestions which should be addressed - none of which are too substantial. Please address these in your rebuttal and manuscript, and if you could again provide a tracked changes version we would be grateful, as this was extremely helpful. We look forward to receiving your finalized paper shortly.

Reviewer comments to Author:

Reviewer: 2

Comments to the Author(s)

The authors have addressed those issues I have previously brought up, and I think the paper will be a valuable addition to the field.

Reviewer: 1

Comments to the Author(s)

Thank you for your careful revisions on the last draft. I have a few more minor suggestions. I have been meticulous on these reviews because I think that suggesting a framework for all scientists to follow is something that should not be taken lightly.

Line 137: add the majority of coral reef studies “from Hughes et al”. These aren’t the majority of coral reef studies, they are the majority from a recent meta-analysis.

Line 152: I am still really having a hard time with ignoring local variability in CO₂/temp in your framework. As the other reviewer stated, local diel variability can be extreme which will have a substantial impact on biological responses. CO₂ is also so different from site to site and even region to region. Even if you don’t have the capability to measure the seawater of the location where you are collecting your biological samples (though this does seem unlikely if you are able to measure the water in an experimental aquarium), then at the very least you can use the extensive data on global ocean CO₂ which has seasonal data by region. I think incorporating this information into your framework is important. Here are a few examples of global ocean CO₂ datasets that are publicly available. Each of these have regional data, and most also have at least seasonal or monthly data.

https://www.nodc.noaa.gov/ocads/oceans/ndp_094/ndp094.html

<https://www.socat.info/>
<https://www.nodc.noaa.gov/ocads/oceans/>

Line 158: There are tons of studies on “in situ biological processes” affecting local CO₂ variation in addition to [31] including, but not limited to:

Kleypas, J. A., Anthony, K. & Gattuso, J. P. Coral reefs modify their seawater carbon chemistry—case study from a barrier reef (Moorea, French Polynesia). *Global Change Biology* 17, 3667–3678 (2011).

DeCarlo, T. M. et al. Community production modulates coral reef pH and the sensitivity of ecosystem calcification to ocean acidification. *Journal of Geophysical Research: Oceans* 122, 745–761 (2017).

Silbiger N.J and Sorte C.J.B. Biophysical feedbacks mediate carbonate chemistry on coastal ecosystems across spatiotemporal gradients *Scientific Reports* 796, (2018).

Page, H. N. et al. Differential modification of seawater carbonate chemistry by major coral reef benthic communities. *Coral Reefs* 35, 1311–1325 (2016).

Smith, J. E., Price, N. N., Nelson, C. E. & Haas, A. F. Coupled changes in oxygen concentration and pH caused by metabolism of benthic coral reef organisms. *Marine Biology* 160, 2437–2447 (2013).

Kapsenberg, L. & Hofmann, G. E. Ocean pH time-series and drivers of variability along the northern Channel Islands, California, USA. *Limnology and Oceanography* 61, 953–968 (2016).

Yeakel, K. L. et al. Shifts in coral reef biogeochemistry and resulting acidification linked to offshore productivity. *Proceedings of the National Academy of Sciences* 112, 14512–14517 (2015).

Krause-Jensen, D. et al. Macroalgae contribute to nested mosaics of pH variability in a subarctic fjord. *Biogeosciences* 12, 4895–4911 (2015).

Similarly, there are many many studies on watershed characteristics and upwelling with respect to affecting carbonate chemistry/temperature regimes. I would add more citations to these statements to highlight the substantial amount of work put into these disciplines.

Units are missing from some of the supplemental files

Author's Response to Decision Letter for (RSOS-191118.R1)

See Appendix B.

Decision letter (RSOS-191118.R2)

04-Dec-2019

Dear Dr Geraldi,

It is a pleasure to accept your manuscript entitled "A framework for experimental scenarios of global change in marine systems using coral reefs as a case study." in its current form for publication in Royal Society Open Science. The comments of the reviewer(s) who reviewed your manuscript are included at the foot of this letter.

on behalf of Prof Kevin Padian (Subject Editor)
openscience@royalsociety.org

Appendix A

We appreciate the opportunity to publish in RSOS and the thoughtful comments by the reviewers which have again improved the manuscript. Please find our responses and actions in **bold** following each of the reviewer's comments.

Reviewers' Comments to Author:

Reviewer: 1

Comments to the Author(s)

I thank the authors for their thoughtful responses and considerations of the prior review. I believe that the authors addressed most of the major concerns by the prior review.

My only major critiques are on figure 2 and the acclimatization section.

First, the authors' response to the reviewer comment on figure 2 was:

"We agree with the reviewer 100% that the relative CO₂ change would be more appropriate and that this varies greatly. However, most studies do not give relative CO₂, but give absolute values. This highlights the issue that most researchers do not know the local CO₂ conditions and variability."

Each of these experiments usually have a "control", "ambient", or "current day" scenario. Even if they report in absolute values the experimental design is that the treatment, CO₂ values are relative to a control. A plot showing delta CO₂ (treatment - control/ambient) will likely get rid of some of this noise, whether or not they know the local variability of the system (in the same way that temperature is presented).

Response: We agree with the reviewer 100% (for real this time), and we apologize for the misunderstanding and not doing this in the previous version.

Action: We updated the Hughes et al 2017 database with the control CO₂, which it did not have. This entailed going through all the CO₂ references (337 entries). We then updated the figures as suggested by the reviewer and provide the updated data in SM 4.

Second, figure 2f assumes that all marine organisms are "losers" of climate change (linear and non-linear responses are all negative and the legend says "marine organisms"). This is an overgeneralization of performance of marine organisms as some species have no or a positive response in different performance measures (see, Kroeker et al. 2013).

Response: This is a very good point by the reviewer.

Action: We have added text to clearly indicated this is just an example showing negative effects although null and positive effects are possible. Verbiage was added on lines 143-144, "Non-linear responses could also result in null effects or positive effects (not shown in Fig. 2f)." and in the figure legend with "The diagram in f represents a negative effect of stressors, but both null and positive effects are also possible."

The “organism acclimatization” section. I completely agree with the authors that understanding how/if organisms can acclimatize to OA/warming is important. However, as currently written, it seems like an afterthought to the main goal of the paper. If acclimatization is brought into the framework then it should be accompanied by suggestions for timing for experiments, etc (as rates of change are brought up in this paragraph). I imagine this is beyond the scope of this study, so I would either remove this section or talk more broadly about how this framework will help us understand responses of marine organisms to climate change. For example, why is acclimatization called out within the context of the framework presented and not species interactions, ecosystem responses, etc.?

Response: We agree with the reviewer that this was beyond the purpose of this manuscript.

Action: The section was removed as suggested

Typo on Line 38: change Rational to rationale

Response: We appreciate the correction

Action: The word was corrected as suggested on line 38.

Reviewer: 2

Comments to the Author(s)

This paper is an interesting examination of experimental conditions used for temperature and CO₂ experiments for corals. While an important topic I feel is either pitched too low for people already in the field who are conducting studies or misses a number of points for those entering the field. The study also duplicates some areas that have been discussed in other papers, e.g. Hughes et al

Response: We appreciate the reviewer’s comments. One of the main reasons for writing this paper was that we wanted to offer researchers a way to choose experimental levels for experiments on both temperature and CO₂, which we think is lacking (as started in lines 41-46) and discussed in the section “Comparison to published experimental suggestions”. We agree that some of the discussion is covered in Hughes et al 2017, as this paper was a major impetus to writing this paper given that our recommendations differ significantly from theirs because we think their recommendations were not based on the best available data nor account for variability in emission projections as stated on line 186-197. Finally, we mention the importance of clearly stating the baseline years, which we think was omitted by Hughes et al. whom used pre-industrial levels for a baseline and not current levels as stated on 123-126. We hope the reviewer can consider these points and perhaps appreciate the contribution that this manuscript could make.

Specific comments

The paper quite rightly advocates for experiments to be put into the context of their local environment, and examines how different oceanic basins are being differentially effected but I think misses variation below this scale and issues with down scaling climate models. For

example Kwiatkowski et al. (Climate Dynamics September 2014, Volume 43, Issue 5–6, pp 1483–1496 | Cite as

What spatial scales are believable for climate model projections of sea surface temperature?) specifically examines the difficulties or using global models to project local (even regional) scale future scenarios. Even smaller than this there is little discussion of the need to include daily variations (this appears to have been brought up in a previous revision but should be further expanded upon). The paper needs to address what scale of local CO₂ and temperature variation is seen, particularly daily changes. In some place temperature can vary but 6y degrees and pCO₂ levels can change over 200 ppm

Response: The reviewer brings up an important point about scale and variation with particular importance to daily variations.

Action: We added an entire paragraph on to discuss this topic on lines 148-155 which states, “We recognize our proposed framework likely oversimplifies the CO₂ and temperature regimes that vary locally, but accurate characterizations of local and regional variability are currently rare, especially for pCO₂ concentrations [30]. As we move forward to characterize and understand drives of high-frequency temperature and CO₂ regimes in coastal systems, this framework could be used to complement baseline observations. This is especially vital for future research focusing on coastal marine habitats that already experience temperatures or pCO₂ levels considerably higher than large-scale means or future projections. Large local variation can result from *in-situ* biological processes [31], watershed characteristics [32] and upwelling [33].”

It is useful to set out best practice but there should also be an acknowledgement of the difficulty in conducting experiments experiments of this type. For example it is not experimentally easy to match temperature and CO₂ conditions, as stated above should this include daily variation. Also the acclimation period. I think all experimentalist would like to acclimate their corals for years but this is logistical almost impossible.

Response: We completely agree that conducting these experiments, particularly with changes in CO₂ are very difficult. We also agree that acclimation is also very difficult.

Actions: We added the following sentence to the conclusion on lines 222-224 to highlight that these experiments are not easy to conduct, “Conducting experiments that replicate local, base-line variation alongside future scenarios necessitates complicated logistical efforts and significant resource investments, especially when dealing with CO₂.”.

As suggested by Reviewer 1, we removed the section on acclimation.

The paper realised heavily on Hughes et al 2017 for their summary of experimental studies,

Firstly this summary is not complete as is missing a number of studies, secondly that is now two years old so I think it is worthwhile including studies from the last two years.

Response: We concur with the reviewer that it would be nice to update this data. However, the primary purpose of this manuscript was to offer a framework for experimental levels. The use of this data was only to illustrate how past studies line up with our framework. Adding the data as suggested would not change our main purpose and thus we would argue is beyond the scope of this manuscript.

Action: We did enhance the data provided by Hughes et al 2017 by revisiting all CO₂ references and adding ambient CO₂ values, which was previously missing.

You need a citation for the sentence, or the analysis in the supps, at line 36 which examines studies on the US west coast.

Response: We appreciate that the reviewer noticed this missing reference.

Action: Reference was added which is now on line 38.

Why is there not RCP2.6 in Figure 2E

Response: We appreciate the reviewer pointing out this omission.

Action: RCP2.6 was added to Fig. 2E

Appendix B

We look forward to seeing this manuscript published in RSOS. We thank the reviewers again for their comments. Below we provide a response and action (if relevant) in bold for each of their comments. Line numbers refer to the version without tracked changes.

Sincerely,
Nathan Geraldi

Reviewer comments to Author:
Reviewer: 2

Comments to the Author(s)

The authors have addressed those issues I have previously brought up, and I think the paper will be a valuable addition to the field.

Response: We thank Reviewer 2 for their time, comments, and approval.

Reviewer: 1

Comments to the Author(s)

Thank you for your careful revisions on the last draft. I have a few more minor suggestions. I have been meticulous on these reviews because I think that suggesting a framework for all scientists to follow is something that should not be taken lightly.

Response: We thank Reviewer 1 for their time, critique, and approval. We agree that suggesting a framework should not be taken lightly and we have done our best to include their suggestions and include all important considerations relevant to these experiments. We also agree that the availability of time-series and local variability is needed. See below for how we address each of the comments.

Line 137: add the majority of coral reef studies “from Hughes et al”. These aren’t the majority of coral reef studies, they are the majority from a recent meta-analysis.

Response: We appreciate this correction.

Action: This was clarified and the sentence on lines 138-139 now reads, “However, the majority of coral reef studies from Hughes et al. [21] (62%) that aimed to simulate future ocean warming and OA manipulated these drivers independently (Fig. 2d).”.

Line 152: I am still really having a hard time with ignoring local variability in CO₂/temp in your framework. As the other reviewer stated, local diel variability can be extreme which will have a substantial impact on biological responses. CO₂ is also so different from site to site and even region to region. Even if you don’t have the capability to measure the seawater of the location where you are collecting your biological samples (though this does seem unlikely if you are able to measure the water in an experimental aquarium), then at the very least you can use the

extensive data on global ocean CO₂ which has seasonal data by region. I think incorporating this information into your framework is important. Here are a few examples of global ocean CO₂ datasets that are publicly available. Each of these have regional data, and most also have at least seasonal or monthly data.

https://www.nodc.noaa.gov/ocads/oceans/ndp_094/ndp094.html

<https://www.socat.info/>

<https://www.nodc.noaa.gov/ocads/oceans/>

Response: We completely agree with the reviewer that local conditions should be used for experimental levels. And that “CO₂ is also so different from site to site and even region to region”. High spatial variability is why we think these large datasets are great but usually do not reflect local conditions needed for choosing levels of experiments (The ocads dataset is great for the open ocean, but does not include coastal areas because they cannot be accurately estimated from global datasets. The socat and nodc are great for existing datasets, but also highlight the limitations, because coastal time-series are limited to << 100 locations over the entire globe).

Action: We state the importance of local variability multiple times, including in the abstract on lines 15-17 with “Although designing experiments in the context of local variability is most appropriate, local temperature and CO₂ characteristics are often unknown as such measures necessitate significant resources, and even less is known about local future scenarios.”, in the introduction on lines 34-35 with “The consideration of local, baseline variation when determining ambient and future experimental levels is optimal [7–9].”, and on 151-153 with “We recognize our proposed framework likely oversimplifies the CO₂ and temperature regimes that vary locally, but accurate characterizations of local and regional variability are currently rare, especially for pCO₂ concentrations [30].”

In addition, we added the two relevant datasets that contain timeseries of pCO₂, so that researchers can check if they are in close proximity to a monitoring station and get an idea of variability for their target levels. The following sentence was added on lines 35-38, “However, local characteristics are often unknown probably because of the significant resources needed to measure them, particularly CO₂ and researchers should be aware of databases that collate relevant datasets (e.g. The Surface Ocean CO₂ Atlas and the Global CO₂ Time-Series and Moorings Project).”.

Line 158: There are tons of studies on “in situ biological processes” affecting local CO₂ variation in addition to [31] including, but not limited to:

Kleypas, J. A., Anthony, K. & Gattuso, J. P. Coral reefs modify their seawater carbon chemistry—case study from a barrier reef (Moorea, French Polynesia). *Global Change Biology* 17, 3667–3678 (2011).

DeCarlo, T. M. et al. Community production modulates coral reef pH and the sensitivity of ecosystem calcification to ocean acidification. *Journal of Geophysical Research: Oceans* 122, 745–761 (2017).

Silbiger N.J and Sorte C.J.B. Biophysical feedbacks mediate carbonate chemistry on coastal ecosystems across spatiotemporal gradients *Scientific Reports* 796, (2018).

Page, H. N. et al. Differential modification of seawater carbonate chemistry by major coral reef benthic communities. *Coral Reefs* 35, 1311–1325 (2016).

Smith, J. E., Price, N. N., Nelson, C. E. & Haas, A. F. Coupled changes in oxygen concentration and pH caused by metabolism of benthic coral reef organisms. *Marine Biology* 160, 2437–2447 (2013).

Kapsenberg, L. & Hofmann, G. E. Ocean pH time-series and drivers of variability along the northern Channel Islands, California, USA. *Limnology and Oceanography* 61, 953–968 (2016).

Yeakel, K. L. et al. Shifts in coral reef biogeochemistry and resulting acidification linked to offshore productivity. *Proceedings of the National Academy of Sciences* 112, 14512–14517 (2015).

Krause-Jensen, D. et al. Macroalgae contribute to nested mosaics of pH variability in a subarctic fjord. *Biogeosciences* 12, 4895–4911 (2015).

Similarly, there are many many studies on watershed characteristics and upwelling with respect to affecting carbonate chemistry/temperature regimes. I would add more citations to these statements to highlight the substantial amount of work put into these disciplines.

Response: We completely agree that there are many studies on both the biological processes and watershed characteristics and how these affect carbonate chemistry. We also agree that we should acknowledge more of these studies.

Action: We have added 3 and 2 additional references with respect to biological and watershed characteristics, respectively and the section on lines 157-158 now reads “Large local variation can result from in-situ biological processes [31–34], watershed characteristics [35–37] and upwelling [38].”

Units are missing from some of the supplemental files

Response: We thank the reviewer for this correction.

Action: Units were added to relevant column titles